# Influence of Magnetizing Conditions on Barkhausen Noise in Fe Soft Magnetic Materials after Thermo-Mechanical Treatment

**DOI:** 10.3390/ma15207239

**Published:** 2022-10-17

**Authors:** Miroslav Neslušan, Katarína Zgútová, Martin Pitoňák, Daniel Kajánek

**Affiliations:** 1Faculty of Mechanical Engineering, University of Žilina, Univerzitná 1, 01026 Žilina, Slovakia; 2Faculty of Civil Engineering, University of Žilina, Univerzitná 1, 01026 Žilina, Slovakia; 3Research Centre, University of Žilina, Univerzitná 1, 01026 Žilina, Slovakia

**Keywords:** Barkhausen noise, low alloyed steels, magnetising voltage, magnetising frequency

## Abstract

Low alloyed steels of low, medium, or high strength are frequently used for many applications in the automotive, civil (bridges), aerospace, and petrochemical industries. A variety of thermomechanical regimes, in which these steels can be produced, enable customization of their matrix with respect to their fatigue resistance, resistance against friction and impact wear, fracture toughness, corrosion resistance, etc. This study analyses the influence of magnetising conditions on Barkhausen noise and other extracted parameters. It was found that the increasing magnetising frequency makes Barkhausen noise weaker, especially in the high strength low alloyed steels, as a result of the decreasing magnetic field in a sample. For this reason, increasing fraction of domain walls is unpinned at the higher frequencies. Barkhausen noise for the high strength low alloyed steels at higher frequencies is remarkably attenuated. Moreover, the different behaviour with respect to direction of the sheet rolling and the transversal direction, can be found due to realignment of the domain walls. This study demonstrates that the position of Barkhausen noise envelopes and the number of Barkhausen noise pulses increase in a systematic manner at the lower magnetising frequencies. Those parameters can be employed for distinction of the low alloyed steels, investigated in this study. However, the increasing magnetising frequency makes attenuation of Barkhausen noise more remarkable for the low alloyed steels of the higher strength. Therefore, the effective value of Barkhausen noise, at the magnetising frequency 750 Hz, in the rolling direction exhibits the systematic descent along with the increasing yield strength. This parameter can be used for distinction of the low alloyed steels after their thermomechanical processing, as well.

## 1. Introduction

Low alloyed steels (LAS) are widely used for many industrial applications because the low-cost materials possess acceptable mechanical and other properties. In particular, the medium or high strength LAS are employed in those cases when the high bearing capacity and the low weight of components are required. The medium strength steels of a fully ferritic matrix are used for the construction of bridges because of their good weldability [1,2]. The high LAS are also integrated into the bearing components in steel and concrete road barriers [3,4,5,6] and in the automotive industry for the car body and wheels, racking systems, truck frames, and various connection profiles [7,8].

The mechanical properties and the corresponding functionality of components strongly depend on the microstructure of LAS, which is linked to the thermomechanical processing as well as the chemical composition [9,10,11]. Increase in the yield and ultimate strength of LAS is usually driven by the hot rolling regime and the consequent cooling rate. Medium LAS are mostly composed of a fully ferritic structure of decreased grain size [10]. The microstructure of high strength LAS is more complex, containing a mixture of ferrite, bainite, and/or martensite [11]. Ti and Nb micro alloying plays a strong role in the strengthening of the LAS matrix. On the one hand, Ti and Nb precipitates increase the density of sites for ferrite nucleation during transformation from the parental austenite to ferrite, which in turn decreases the matrix grain size. The Ti and Nb precipitates hinder dislocation motion and also contribute to the higher yield and ultimate strength. Finally, a certain volume of Nb and Ti is directly dissolved in the Fe matrix [10,11]. The high strength of LAS is compensated by the decreased formability and limited strain hardening.

Components in the automotive industry made of LAS are produced in thousands and millions of pieces, and their state and the corresponding functionality can vary. Moreover, civil constructions made of LAS are exposed to stress and corrosion attack. Therefore, non-destructive techniques are very often employed for such purposes. Magnetic Barkhausen noise (MBN) represents a very fast and reliable technique for the characterisation of ferromagnetic bodies. The magnetic substructure of the domains and the corresponding positions of the domain walls (DWs) are altered under mechanical load and/or altering magnetic field. The motion of the DWs is irreversible and discontinuous, producing electromagnetic pulses well known as MBN [12,13,14]. DWs in their motion encounter all lattice imperfection. Therefore, MBN contains information about the microstructure. Dychtoń et al. [15] linked increasing MBN with thermal softening after grinding and remarkable alteration of hardness in the thermally affected region. Neslušan et al. [16] investigated that the lower MBN after grinding can be found when carbides in martensite are unaffected and their decomposition is observed when the MBN exceeds the critical threshold. Stupakov et al. [17] reported that the stronger MBN emission is received due to the presence of decarburised surface layer of low dislocation density. Ktena et al. [18] found that the MBN is increasing due to grain refinement, as a result of increasing DWs density. Furthermore, the alignment of the DWs is affected by the stress state, and therefore, MBN can be employed for the measurement of stresses. Liu et al. [19] clearly proved that the MBN grows along with tensile stresses, as a result of the DWs realignment. Kypris et al. [20] provided the model in which the residual stress assessment can be obtained using MBN depth attenuation. Sorsa et al. [21] developed the regression model for prediction of residual stresses after the shot peening.

The MBN technique has already been used for monitoring LAS [22,23] or assessing stresses in real highway bridges [6]. Pitoňák et al. [22] reported that MBN attenuates in LAS as results of variable Zn coating. Jančula et al. [23] found that MBN decreases along with increasing corrosion extent in LAS. Zgútová et al. [6] reported on sensitivity of different MBN parameters with respect to tensile stress in LAS S460. Moreover, MBN as a function of strain hardening of S235 and tensile stress was investigated as well [24,25]. Neslušan et al. [24] clearly demonstrated that the MBN is decreasing in the direction of tensile stress after the uniaxial plastic straining, whereas Jurkovič et al. [25] found remarkable non-homogeneity of plastic straining and the corresponding MBN after yielding. Blaow et al. [26] investigated the effect of LAS bending on MBN and reported about the decreasing MBN in the region of compressive stresses. Gutierez et al. [27] demonstrated the decreasing MBN as a result of reduced hardness and the corresponding dislocation density in AISI 4130. Kadavath et al. [28] also clearly proved, using a Jominy end-quenched test and the samples subjected to different cooling rates, that the MBN decreases with the increasing hardness. Saleem et al. [29] studied temper embrittlement of LAS with MBN and explained the altered MBN signals in terms of the impurity migration effect.

It is worth mentioning that MBN is not only a function of the microstructure and/or stress state, but the magnetising conditions also play a significant role [30,31]. In particular, the magnetising voltage (and the corresponding magnetic field) and frequency strongly affect the fraction of movable DWs and their speed of motion. Therefore, the influence of the magnetising conditions on MBN (and extracted MBN features) should be carefully investigated.

The MBN signal post-processing enables extraction of different MBN parameters, which can be employed for a proposal of suitable concepts when materials characterisation is required in a non-destructive manner. The effective value of MBN signal is a function of the number of detected pulses, as well as their strength. Those pulses can occur in different ranges of magnetic fields, with the different distribution. For this reason, the MBN envelopes can be also employed for extraction of the peak position as the maxima of the MBN envelopes, together with the width of these envelopes. The aforementioned MBN parameters are considered as the most common MBN features, frequently integrated into scientific studies dealing with the MBN (easily extracted, mostly provided by commercially available software) [31]. Those parameters can also be linked with the physical processes associated with the DWs motion, which provides deeper insight into magnetisation of bodies [18,22,31].

Studies, dealing with the influence of magnetising conditions and usage of the different MBN parameters for a specific purpose, have been reported. The procedures for setting the optimised magnetising voltage and frequency can be found in [30]. Vashista and Moorthy [31] reported that the alterations in magnetic field affect appearance of MBN envelopes. It was also demonstrated that the position of an MBN envelope can be easily linked with dislocation density and the corresponding hardness [32]. However, Blažek et al. [33] found that the MBN envelope is also sensitive to the MBN signal filtration, especially at the higher magnetising frequencies. Santa-aho et al. [34] employed the variable magnetising sweeps for detection of the case-hardened depth. Santa-aho et al. [35] also executed features applicability such as MBN, MBN envelopes and other features for prediction of the residual stresses in the case-hardened bodies. Šrámek et al. [36] found that the width of an MBN envelope drops along with the increasing magnitude of tensile stresses. In addition, the number of MBN pulses was employed for deeper insight into magnetisation process of LAS. Jurkovič et al. [25] found that the refinement of microstructure after plastic deformation increases the number of detected pulses. These studies proved that proposal of suitable magnetising conditions is of a vital importance in order to obtain acceptable sensitivity of the MBN technique. Having in mind the diversity of LAS as well as a variety of magnetising conditions, multiple combinations of optimal magnetising conditions for LAS of different yield strengths can be found. This paper investigates this aspect in a systematic manner and explains the influence of the magnetising conditions on the acquired MBN signals. As compared to the previous studies, this study provides complex and systematic investigation in the field of LAS, when the altered magnetic anisotropy is combined with the variable yield strength. Furthermore, this study also lists the suitable MBN parameters which can be easily employed for reliable distinction between the LASs.

## 2. Materials and Methods

The experimental study was carried out on commercially available LAS of variable yield strength, as listed in Table 1 (the number in the first column refers to the nominal yield strength of the LAS). The samples for the measurements were cut off of sheets of 5 mm thickness. The mechanical properties as well as the MBN signals were measured along the rolling direction (RD) of the sheets as well as in the transversal direction (TD).

The samples of 70 mm length and 30 mm width were used for the measurements. The initial screening of the delivered sheets revealed that the near surface layer of about 0.1 mm thickness contains a thermally softened region of rough grains and lower hardness as contrasted against the volumetric matrix. For this reason, this layer was electrolytically etched off. The detailed and complex information about the microstructure, mechanical properties, and EBSD observation of the measurements of the magnetic parameters can be found in the technical report [32]. However, the main findings can be briefly summarized because of their importance with respect to the true interpretation of the MBN signals.

The microstructure of S235, S355, and MC500 is fully ferritic, and the grain size decreases with the yield strength, see Figure 1. The microstructure of MC700 is a mixture of ferrite and bainite, and the microstructure of MC960 is a mixture of bainite and martensite, whereas MC1100 is mostly composed of martensite and limited fraction of bainite, see also Figure 1. The microstructure is a product of hot rolling and the consequent cooling rate. The details about the thermomechanical processing are not known, but it is considered that the degree of sheet reduction during hot rolling is increasing and the cooling is accelerated along with the increasing yield strength. Grain refinement in the case of fully ferritic steels is connected with the increasing tensile stresses, which tends to be released at accelerated cooling rates. The hardness *HV1* increases with the nominal yield strength of LAS as a result of the increased dislocation density. The true yield and ultimate strength are more as the nominal ones and increasing yield strength is compensated by the decreasing elongation at break [32].

MBN measurements were carried out using RollScan 350 (serial sensor S1-18-12-01, Stresstech, Jyväskylä, Finland). Voltage and frequency sweeps were measured using the ViewScan software (version V.5.4b (2015-0-06) [30] and represent the MBN in the frequency range from 70 to 200 kHz. MBN signal acquisition was carried out using the software MicroScan–recording signal, its filtration (10 to 1000 kHz), extraction of MBN parameters such as the effective value of the signal (referred to as MBN), *PP* (Peak Position that refers to the maxima of the MBN envelope), *FWHM* (Full width at half maxima of the MBN envelope), as well as number of detected pulses. This software also provides information about the MBN envelopes and FFT spectra of the MBN. In order to subtract the MBN background noise produced by the serial sensor [33], the MBN signals were exported and processed in the self-made software in order to calculate the effective value of the MBN and the number of MBN pulses below the background threshold (using the MBN strength threshold of 100 mV).

MBN as well as the sweeps were measured in the RD and TD. Based on the voltage MBN sweeps (indicated in the next chapter), a magnetising voltage of 5 V was employed (such voltage corresponds to the altering magnetic field of amplitude 4.63 kA.m^−1^) and kept constant for all magnetising frequencies, i.e., 125, 250, 500, and 750 Hz. All extracted MBN parameters were obtained by averaging the values from 6 consecutive bursts.

## 3. Results of Experiments and Their Discussion

### 3.1. Voltage and Frequency Sweeps

A suitable magnetising voltage and frequency can be adjusted using the magnetising sweeps [30,34]. This routine is involved in the RollScan 350 device and can be recorded using the ViewScan software. The manufacturer recommends the adjustment of the magnetising voltage at the knee of the voltage sweeps, as illustrated in Figure 2a,b. This way, the optimised sensitivity of the MBN technique can be attained. Proposal of the optimised voltage for one material of low magnetic anisotropy is therefore easy task. As soon as, the different ferromagnetic bodies of the different magnetic anisotropies (expressed in terms of MBN measured in the different directions) are considered a certain balance (compromise) with respect to the different evolution of voltage sweeps should be taken into account.

Figure 2a,b illustrate that the knee for all LAS can be found roughly at 5 V in the RD as well as the TD (only the voltage sweep for MC1100 in the RD is different). Before this threshold, the magnetising sweep shows an ascending behaviour because the increasing strength of the magnetising field increases the fraction of movable DWs contributing to the MBN. Beyond this threshold, the voltage sweeps saturate and no further valuable increase can be detected. Expressed in other words, the magnetising field of amplitude ± 4.63 kA.m^−1^ produced by the sensor is strong enough to unpin a majority of the DWs in the matrix for all LAS (apart from MC1000 in the RD). The difference in the optimal magnetising voltage is minor. For this reason, a magnetising voltage of 5 V was kept constant when the magnetising frequency was altered.

As contrasted against the voltage sweeps, the evolution of the frequency sweeps (see Figure 2c,d) is quite different with respect to the different LAS as well as with respect to the RD and TD. The evolution of the frequency sweep is typical for those obtained from the RollScan sensing system when the ascending region at the lower frequencies is replaced by the descending region at the higher frequencies. The optimal magnetising frequency should be selected at the maximum of the frequency sweep in order to obtain the highest MBN and, therefore, the highest sensitivity of the MBN technique. However, this maximum is different for the different LAS (see Figure 3) and valuably decreases with the increasing nominal strength of the LAS. The influence of the RD and TD is less pronounced. The maximum of the frequency sweep in the RD is comparable up to the nominal yield strength of 700 MPa, followed by a medium drop for MC960 and a remarkable decrease for MC1100. The differences in the maximum in the TD are less remarkable. The rate of descent beyond the maximum of the frequency sweep is only moderate for S235 and becomes steeper along with the increasing yield strength of the LAS. This statement is valid for the RD as well as the TD.

Alteration of the magnetising frequency plays a major role in the measured MBN as contrasted against the magnetising voltage. Two aspects of the magnetising conditions should be considered. The first one is associated with the amplitude of the magnetising field *H*, whereas the second one is associated with its rate of change in time *d**H*/*d**t* (can be easily calculated from the known amplitude of the magnetising field and its frequency). The evolution of magnetisation *M* is closely related to *H*, see Equation (1). Figure 4a demonstrates that *H* as well as *d**H*/*d**t* grow with the magnetising voltage, which in turn corresponds to the monotonous growth of the voltage sweeps (or their saturation at the higher voltages when all DWs are unpinned). Expressed in other words, the superimposing effect of *H* as well as d*H*/d*t* contribute to the growing MBN along with the magnetising voltage. The increasing MBN with the magnetising voltage (see Figure 2a,b) is also linked with the increasing rate of magnetisation change due to the increasing DWs speed of irreversible motion *v_p_* (see Equation (1)).
(1)$$v_{p}=\frac{dM}{dt}={\left(H-H_{c}\right)}^{q}\;(m.s^{-1})$$
where *M* is magnetisation, *H* magnetic field, *H_c_* the magnetic field for unpinning DWs and *q* is the exponent (≈0.5) [37]. Having the constant *H_c_*, the higher *H* at the higher magnetising voltage accelerates *v_p_* of DWs in motion due to increasing difference between *H* and *H_c_*.

The situation with respect to the magnetising frequency is more complicated. The amplitude of the magnetising field *H* drops down with the magnetising frequency together with increasing *d**H*/*d**t* (see Figure 4b). For this reason, MBN grows with respect to the increasing *d**H*/*d**t* compensated by the decreasing amplitude of the magnetising field *H*. Being so, the contribution of the increasing *d**H*/*d**t* dominates over the decreasing amplitude of *H* at the lower magnetising frequencies, whereas the influence of the weak *H* prevails at higher magnetising frequencies. The steeper descent at the higher magnetising frequencies for LAS of higher strength is due to the higher pinning strength of the matrix containing a higher dislocation density expressed in the term *HV1* (indicated in Table 1). The decreasing *H* at the higher magnetising frequencies makes lower *H − H_c_*. As soon as *H − H_c_* is negative, DWs are pinned in their positions. This effect becomes more pronounced for the LAS of higher strength due to the higher dislocation density linked with *HV1* (see Table 1). Expressed in other words, the higher sensitivity MBN against the magnetising frequency can be found along with the increasing opposition of the pinning sites (as well as their increasing density) when lower magnetising fields are not capable of initiating the irreversible motion of the DWs.

One might argue that MBN dominates the nearby coercive force (nearby the zero magnetic field and the corresponding magnetising current) when the magnetic field is low [18,24,26], whereas the DW rotation dominates at a higher magnetic field [12,13,14], see Figure 5. For this reason, the amplitude of the magnetising field should not influence the MBN. However, this part of the study demonstrates that a certain fraction of DWs at lower magnetising fields really stays unpinned and *d**H*/*d**t* only affects the speed of the movable DWs [37].

### 3.2. MBN Parameters as a Function of the Magnetising Frequency

The increase in MBN in the RD at the lower magnetising frequencies (see Figure 6a) is due to the ferrite grain refinement as a result of the decreasing grain size and the corresponding growth in the DW density [32]. As soon as the grain refinement is replaced by the phase transition (bainite + ferrite for MC700), the moderate descent can be found as a result of the DW realignment into the TD direction [24,32]. This realignment is driven by increasing strain hardening [9,10] during hot forming. The final steep decrease in MBN in the RD is due to the high dislocation density and strong opposition of pinning sites against the DW motion (valid for the RD as well as the TD). MBN in the RD for the magnetising frequency of 750 Hz exhibits a monotonous and progressive decrease as a result of the reduced number of movable DWs as aforementioned (due to the weak altering magnetic field). The progressive decrease in MBN for LAS of higher nominal yield strength is driven by the synergistic contribution of the DW realignment into the TD and the lower fraction of movable DWs. The evolution of MBN in the RD for the magnetising frequency of 750 Hz can be used as the distinct parameter for easy materials separation because MBN continuously drops down along with the nominal yield strength as compared with the others magnetising frequencies (or TD).

On the other hand, MBN in the TD exhibits continuous growth at the lower magnetising frequencies. MBN for the fully ferritic body increases due to the increasing DW density. MBN for the LAS of higher nominal yield strength is also due to the superimposing DW realignment, as mentioned above. It should also be noted that the evolution of the MBN, for the magnetising frequency of 750 Hz, differs from those for the lower magnetising frequencies in RD as well TD, see Figure 6. The MBN systematically decreases at the lowest magnetising frequency of 750 Hz (see Figure 6a) since the fraction of DWs, which are pinned in their position, increases along with the increasing yield strength (additionally, the superimposing contribution of DWs realignment into TD takes a certain role). On the other hand, it seems that the major fraction of the DWs is unpinned and contributes to the MBN at the lower magnetising frequencies. The descending region, for the magnetising frequency of 750 Hz in TD, is followed by the moderate growth of the yield strength for the LAS, in the range from 500 to 960 MPa, since the decreasing fraction of DWs in motion is compensated by the DWs realignment into TD (see Figure 6b).

MBN envelopes (see Figure 7) and the corresponding *PP* positions (see Figure 8) exhibit very good sensitivity against the increasing dislocation density expressed in terms of *HV1*. *PP* continuously grows along with the nominal yield strength and *HV1* (see also Table 1). The best sensitivity exhibits the lowest magnetising frequency of 125 Hz. The evolutions of *PP* versus the nominal yield strength tend to flatten when the magnetising frequency is higher. Expressed in other words, the sensitivity of the *PP* parameters with respect to the materials selection is decreasing when the magnetising frequency is increasing.

*PP* in the RD for LAS of lower nominal yield strength is lower at the lower magnetising frequencies, but this evolution is reversed for MC1100, see Figure 7. This behaviour indicates the strong role of the pinning strength of pinning sites with respect to the altered magnetic field. The decreasing magnetic field for LAS of lower nominal yield strength has only a minor role in the decreasing MBN because the pinning strength of the pinning sites is low. As soon as the pinning strength grows, the decreasing magnetic field is not capable of unpinning some DWs—especially those pinned by the sites of higher pinning strength. For this reason, MBN as well as the height of the MBN envelopes decrease, contains the MBN pulses from the DWs being weakly pinned only. That is the reason why the MBN envelopes are shifted to the lower magnetic fields (see Figure 8b). Figure 8 also illustrates that the *PP* is nearly unaffected by the magnetising frequency for the LAS of the yield strength of 960 MPa, so this LAS represents the limit when the ascending evolution of the *PP* versus magnetising frequency is reversed by the increasing one. In addition, Figure 8b shows that the *PP* for this LAS are quite close. It can be therefore noted that the effects of the movable DWs fraction and their dynamics are balanced in this particular case.

That is also the reason why the *FWHM* of the MBN envelope drops down at higher magnetising frequencies, see Figure 9 This parameter usually indicates the range of magnetic fields in which MBN can be detected. The decreasing *FWHM* clearly proves that a certain fraction (unpinned DWs) of MBN pulses is missing and MBN occurs in the narrower range of magnetic fields. This is the reason why the number of detected MBN pulses decreases with the magnetising frequency, see Figure 10.

On the other hand, the evolution of all detected pulses (ascending with respect to the nominal yield strength) and those exceeding the threshold of 100 mV (descending with the nominal yield strength) is different and demonstrates that the density of DWs for LAS of higher nominal strength is higher [32]. However, the strength of the MBN pulses originating from the denser matrix of the DWs is lower (very often, these pulses are below the background threshold of 100 mV).

Table 2 summarizes the usability of the different MBN parameters in order to distinguish among the LAS of variable yield strength. Only the effective value of MBN at the magnetising frequency 750 Hz and the number of MBN pulses for the magnetising frequency 250 Hz can be used for the aforementioned purpose (highlighted by red colour). On the other hand, *PP* and the number of MBN pulses highlighted by blue colour can be employed as their combination since in some cases the neighbouring points in the evolutions are similar, see Figure 8 and Figure 10.

The influence of the magnetising frequency on MBN can also be demonstrated on the FFT spectra of the detected signals, as illustrated in Figure 11. The FFT spectra for S235 in the RD, TD, as well all magnetising frequencies are more or less similar, and this body is less sensitive to the proposal of suitable magnetising conditions (see Figure 11a,b). As soon as the strength of the pinning sites and their density increases and a remarkable magnetic anisotropy due to the DWs realignment into the TD takes place, the remarkable difference in MBN and the corresponding FFT spectra can be obtained (see especially Figure 11c,d).

Finally, the influence of skin depth *δ* should be noticed. Skin depth can be calculated as follows:(2)$$\delta =503\;\sqrt{\frac{\rho }{\mu_{r}.f}}\;(\mathrm{mm})$$
where *f* is the magnetising frequency, *μ_r_* is the permeability, and *ρ* is the resistivity of the medium [12,13,14]. Skin depth calculated by the use of Equation (2) refers to the penetration depth of the altered magnetic field. On the other hand, *δ* can affects the reading depth since *δ* can affect the magnetic flux density.

Having the resistivity of the medium 9.71 × 10^−8^ S.m^−1^ nearly the same for all investigated steels, and information about permeability reported in [32] (see also Figure 12a) m the skin depth can be easily calculated as a function of the magnetising frequency and/or the yield strength, see Figure 12b. The skin depth of the bodies of lower permeability makes *δ* higher which in turn decreases the true intensity of *H* within the reading depth. This effect contributes to the higher MBN since increases *H* − *H_c_*. However, Figure 12 clearly demonstrates that the differences in permeability are too low which in turn makes the similar skin depths for all LAS. For this reason, this effect does not play any role in MBN as well as the evolutions of parameters extracted from the raw MBN signals.

## 4. Conclusions

The main outputs of this study can be briefly summarized as follows:-The altering magnetising voltage has only a minor role with respect to the MBN emission, and a magnetising voltage of about 5 V is strong enough to unpin the majority of DWs for nearly all investigated LAS;-The altering magnetising frequency plays a major role, and especially LAS of higher nominal yield strength are very sensitive to the weakening of the altering magnetic field;-The weakening of the magnetic fields at the higher magnetising frequencies causes that certain fraction of DWs is unpinned (especially in the case of LAS of higher nominal yield strength), which in turn decreases the *FWHM* of the MBN envelopes, decreases the sensitivity of *PP* for the differentiation of LAS, and reduces the number of detected pulses;-LASs of the lower nominal yield strength are less sensitive to the proposal of suitable magnetising conditions, whereas the remarkable difference in MBN can be obtained with respect to the altered magnetising frequency as well as direction of measurements for LAS of higher nominal yield strength.

Finally, the role of MBN for monitoring LAS should be discussed. The optimal magnetising conditions can be easily proposed using the voltage and frequency sweeps when multiple components of the same LAS are monitored. As soon as the MBN technique is employed to distinguish among the LAS of different nominal yield strength, the suitable MBN parameters as well as magnetising conditions should be reconsidered.

Applicability of different MBN parameters, for distinction of the LASs of the different yield strength, can be summarized as follows:-MBN in the RD at the magnetising frequency of 750 Hz exhibits the systematic descent (90% decrease) and can be considered as the suitable MBN parameter, employed for this purpose (whereas the MBN in RD at the lower magnetising frequencies, as well as in TD, behave in a non-systematic manner);-*PP* in the RD and TD at the lower magnetising frequencies (125 Hz and 250 Hz) are increasing along with the yield strength of LAS (50% increase for 125 Hz in RD and 40% increase in TD, 23% increase for 250 Hz in RD and 16% increase in TD); however, the *PP* should be combined with the number of MBN pulses (since at the higher magnetising frequencies the *PP* behave in a non-systematic manner);-*FWHM* behaves in a non-systematic manner for all the magnetising conditions and directions;-the number of all the MBN pulses, at the lower magnetising frequencies (especially 250 Hz), is increasing along with the yield strength of LAS (25% increase for 125 Hz in RD and 23% increase in TD, 37% increase for 250 Hz in RD and 30% increase in TD); however, the number of MBN pulses should be combined with the *PP* for the magnetising frequency of 125 Hz (the number of the MBN pulses at the higher magnetising frequencies behave in a non-systematic manner);-applying the background threshold for counting of the MBN pulses number is not beneficial, in this particular case, with respect to materials characterisation.

## Figures and Tables

**Figure 1 materials-15-07239-f001:**
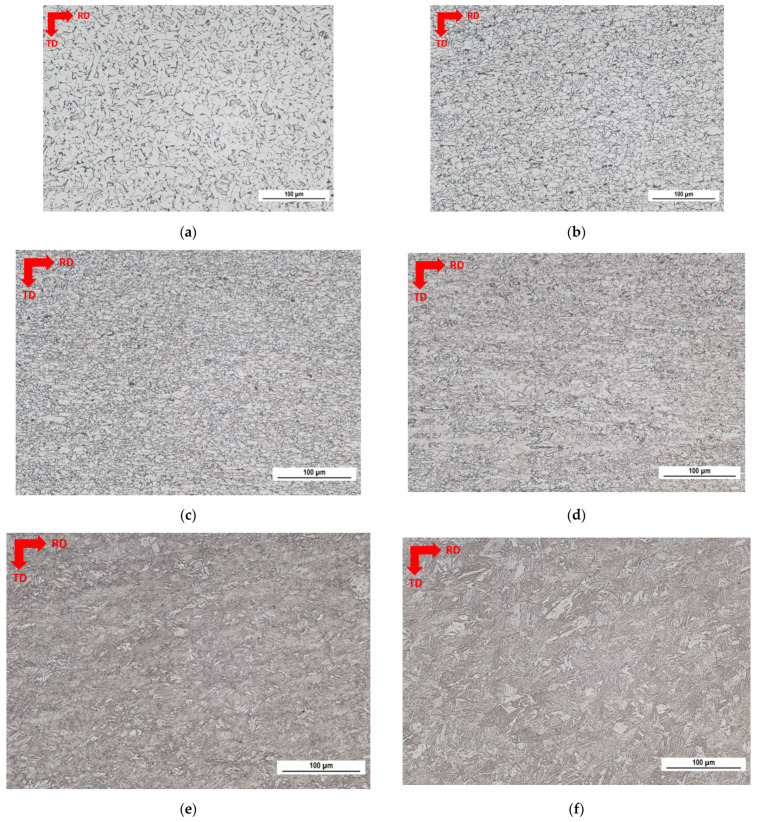
Metallographic images of investigated steels (scanned area approx. 0.238 mm^2^). (**a**) Nominal *σ_YS_* = 235 MPa, (**b**) nominal *σ_YS_* = 355 MPa, (**c**) nominal *σ_YS_* = 500 MPa, (**d**) nominal *σ_YS_* = 700 MPa, (**e**) nominal *σ_YS_* = 960 MPa, (**f**) nominal *σ_YS_* = 1100 MPa.

**Figure 2 materials-15-07239-f002:**
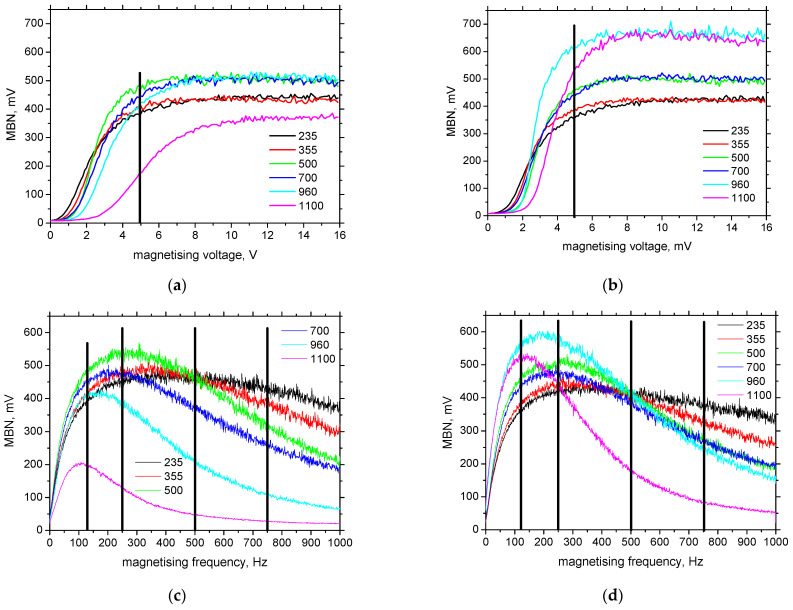
Voltage and frequency sweeps for the LAS of variable yield strength. (**a**) Voltage sweep in the RD—magnetising frequency of 125 Hz, (**b**) voltage sweep in the TD—magnetising frequency of 125 Hz, (**c**) frequency sweep in the RD—magnetising voltage of 5 V, (**d**) frequency sweep in the TD—magnetising voltage of 5 V.

**Figure 3 materials-15-07239-f003:**
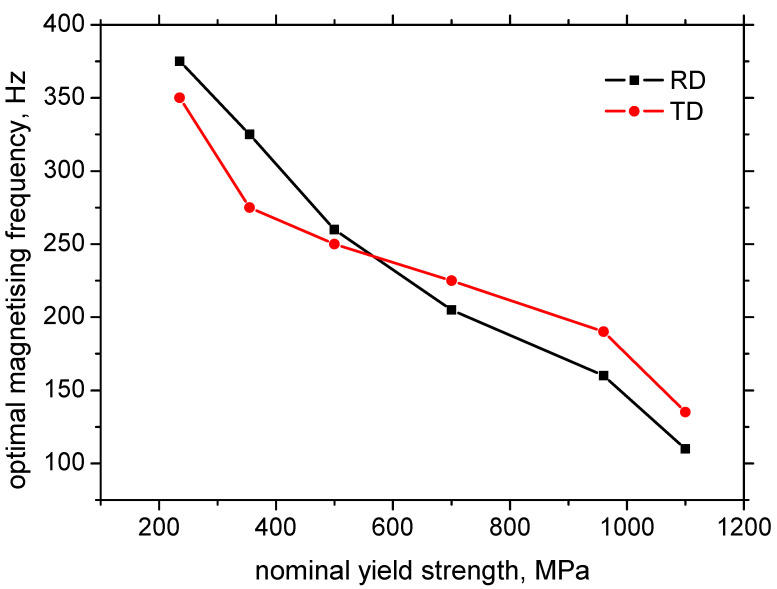
Optimal magnetising frequency as a function of the nominal yield strength.

**Figure 4 materials-15-07239-f004:**
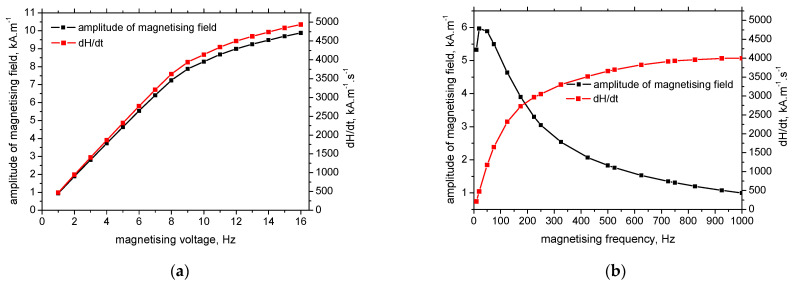
Amplitude of the magnetic field and *d**H*/*d**t* as a function of the magnetising conditions. (**a**) Amplitude of the magnetic field and *d**H*/*d**t* as a function of the magnetising voltage. (**b**) Amplitude of the magnetic field and *dH*/*dt* as a function of the magnetising frequency.

**Figure 5 materials-15-07239-f005:**
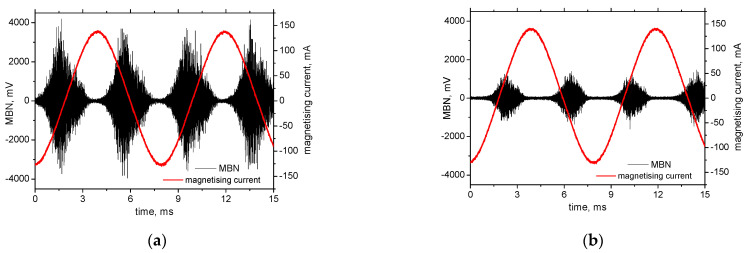
MBN signal in the RD, magnetizing frequency of 125 Hz. (**a**) S235, (**b**) MC1100. Amplitude of the magnetising current of 140 mA corresponds to the amplitude of the magnetic field of 4.63 kA.m^−1^.

**Figure 6 materials-15-07239-f006:**
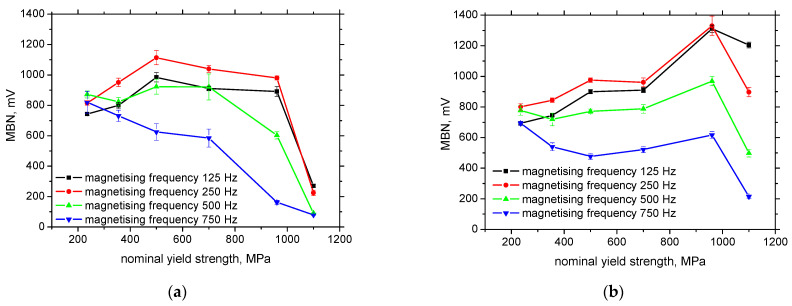
MBN as a function of the nominal yield strength. (**a**) RD, (**b**) TD.

**Figure 7 materials-15-07239-f007:**
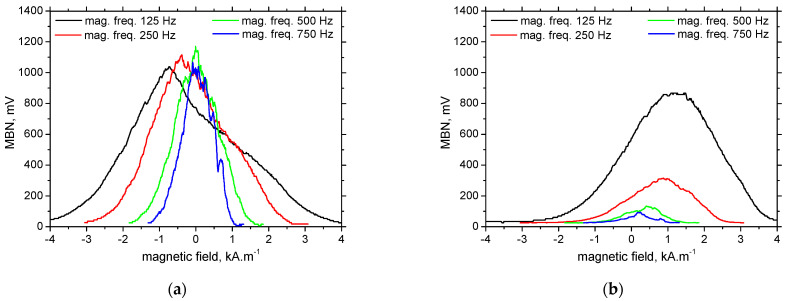
MBN envelope as a function of the magnetising frequency, RD. (**a**) S235, (**b**) MC1100.

**Figure 8 materials-15-07239-f008:**
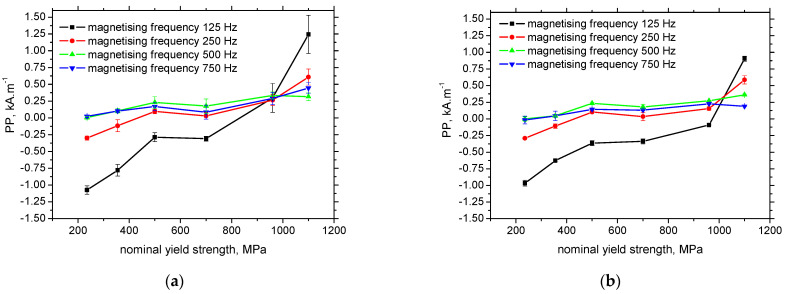
*PP* as a function of the nominal yield strength. (**a**) RD, (**b**) TD.

**Figure 9 materials-15-07239-f009:**
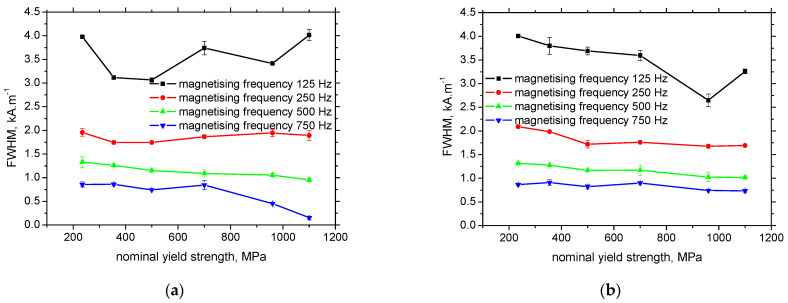
*FWHM* as a function of the nominal yield strength. (**a**) RD, (**b**) TD.

**Figure 10 materials-15-07239-f010:**
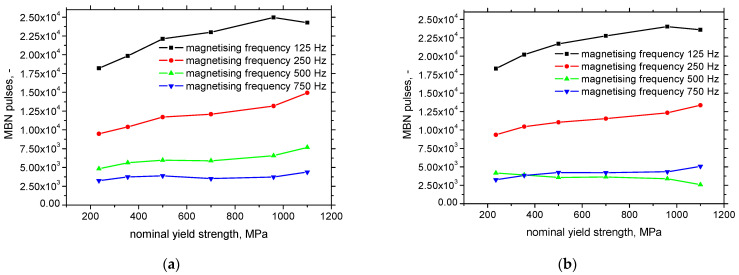
Number of MBN pulses as a function of the nominal yield strength. (**a**) All detected pulses—RD, (**b**) all detected pulses—TD, (**c**) number of pulses above threshold 100 mV—RD, (**d**) number of pulses above threshold 100 mV—TD.

**Figure 11 materials-15-07239-f011:**
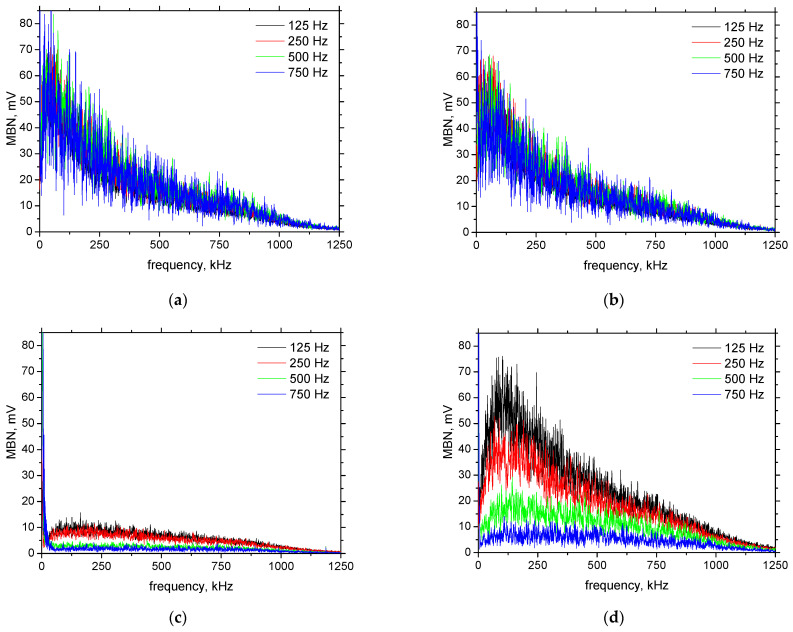
FFT spectrums of MBN signals. (**a**) S235-RD, (**b**) S235-TD, (**c**) MC1100-RD, (**d**) MC1100-TD.

**Figure 12 materials-15-07239-f012:**
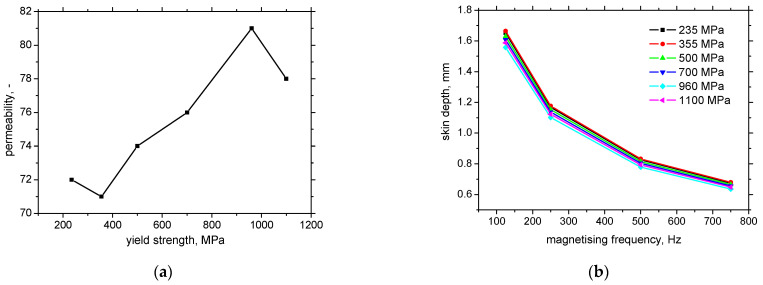
Permeability and skin depth as a function of yield strength and magnetising frequency. (**a**) permeability, (**b**) skin depth as a function of magnetising frequency.

**Table 1 materials-15-07239-t001:** List of investigated LAS, their microstructure, residual stresses, chemical composition, and *HV1*.

LAS	Microstructure	Residual Stresses, MPa		Chemical Composition, Weight %				*HV1*
		RD	TD	C	Mn	Si	Nb + Ti	
S235	Ferrite	−45 ± 33	−38 ± 32	0.22	1.6	0.05	-	125 ± 8
S355	Ferrite	147 ± 15	133 ± 18	0.07	0.7	0.01	0.05	158 ± 2
MC500	Ferrite	315 ± 7	256 ± 18	0.08	1.1	0.02	0.1	214 ± 5
MC700	Ferrite + bainite	211 ± 40	126 ± 20	0.05	1.8	0.02	0.21	265 ± 5
MC960	Bainite + martensite	21 ± 12	25 ± 6	0.12	1.3	0.25	0.12	362 ± 16
MC1100	Martensite + bainite	−27 ± 18	30 ± 5	0.15	1.8	0.5	-	432 ± 14

**Table 2 materials-15-07239-t002:** List of MBN parameters and their usability to distinguish among the LAS of the different yield strength.

	*rms,* mV		*PP,* kA.m^−1^		*FWHM,* kA.m^−1^		MBN Pulses-All		MBN Pulses–above 100 mV	
	RD	TD	RD	TD	RD	TD	RD	TD	RD	TD
125 Hz	x	x	+50%	+40%	x	x	+25%	+23%	x	x
250 Hz	x	x	+23%	+16%	x	x	+37%	+30%	x	x
500 Hz	x	x	↑?	↑?	↓?	↓?	x	x	x	x
750 Hz	- 90%	x	↑?	↑?	x	x	x	x	x	x

x—non-systematic evolution of MBN parameter. - number—descending evolution (capable to distinguish among the LAS as the sole MBN parameter). + number—ascending evolution (capable to distinguish among the LAS as the sole MBN parameter). + number—ascending evolution (only as the combination of PP and the number of all MBN pulses) ↓?—too flat descending evolution (the descend usually within the standard deviation) ↑?—too flat ascending evolution (the ascent usually within the standard deviation).

## Data Availability

The raw data required to reproduce these findings cannot be shared easily due to technical limitations (some files are too large). However, authors can share the data on any individual request (please contact the corresponding author by the use of their mailing address).

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
