# Peer review of "Influence of Magnetizing Conditions on Barkhausen Noise in Fe Soft Magnetic Materials after Thermo-Mechanical Treatment"

_materials, 2022, doi:10.3390/ma15207239_

Round 1

Reviewer 1 Report

The authors presented an article about “effect of magnetizing conditions on Barkhausen noise in ferro soft magnetic materials. It was made after thermos mechanical treatment. I think the paper appropriate for materials journal but, it should be revised in order to be ready for publication.

1.      Demonstrate in the abstract novelty, practical significance. Abstract should be expanded sentences related to the results. The results of the study should be given as numerical percentages.

2.      The introduction part is well-organized yet there is a reference problem. First, your reference list contains a few papers from polymers journal. If your work is convenient for this journal’s context then there are many references from this journal. Secondly, cited sources should be primary ones. Namely, indexed area shows the power of a paper and directly your paper’s reliability. Please make regulations in this direction.

3.      It is necessary to give quantitative and qualitative indicators of the proposed method in conclusion. Conclusions should be written in more detail adding numeric data. Conclusions section is inadequate. There should be the importance of the study in detail, comparison results with other approaches in literature, the success of the prediction and computational results.

4.      Improve the results and discussion and conclusion parts.

5.      There are typographical and eventual language problems in paper

6.      It should be clearly stated the success rate of the study in abstract, results and discussion and conclusion parts.

Author Response

Reviewer n. 1:

All changes made in the manuscript (additional texts and corrections) are highlighted yellow colour (valid for the manuscript as well as this document).

Reviewer: Demonstrate in the abstract novelty, practical significance. Abstract should be expanded sentences related to the results. The results of the study should be given as numerical percentages.

Response: We altered the abstract as well as added information about the novelty. The numbers were added in Table 2.

Manuscript: please check the appearance of Table 2

Abstract

… behaviour with respect to the direction of sheet rolling and the transversal direction can be found due to realignment of the domain walls. This study demonstrates that the position of Barkhausen noise envelopes and the number of Barkhausen noise pulses increase in a systematic manner at the lower magnetising frequencies and these parameters can be employed for distinction of the low alloyed steels investigated in this study. However, the increasing magnetising frequency makes attenuation of Barkhausen noise more remarkable for the low alloyed steels of the higher strength. Therefore, the effective value of Barkhausen noise at the magnetising frequency 750 Hz in the rolling direction exhibits the systematic descent along with the increasing yield strength and this parameter can be also used for distinction of the low alloyed steels after their thermomechanical processing. 

Novelty

Studies dealing with the influence of magnetising conditions and usage of the different MBN parameters for a specific purpose have been reported. The procedures for setting the optimised magnetising voltage and frequency can be found in [30]. Vashista and Moorthy [31] reported that the alterations in magnetic field affect appearance of MBN envelopes. It was also demonstrated that the position of MBN envelope can be easily linked with dislocation density and the corresponding hardness [32]. However, Blažek et al. [33] found that MBN envelope is also sensitive to the MBN signal filtration especially at the higher magnetising frequencies. Santa-aho et al. [34] employed the variable magnetising sweeps for detection of case-hardened depth. Santa-aho et al. [35] also executed features applicability such MBN, MBN envelopes and other features for prediction residual stresses in case-hardened bodies. Šrámek et al. [36] found that the width of MBN envelope drops down along with the increasing magnitude of tensile stresses. Also the number of MBN pulses was employed for deeper insight into magnetisation process of LAS. Jurkovič et al. [25] found that the refinement of microstructure after plastic deformation increases the number of detected pulses. These studies proved that proposal of suitable magnetising conditions is vital of importance in order to obtain acceptable sensitivity of MBN technique. Having in mind the diversity of LAS as well as a variety of magnetising conditions, multiple combinations of optimal magnetising conditions for LAS of different yield strengths can be found. This paper investigates this aspect in a systematic manner and explains the influence of the magnetising conditions on the acquired MBN signals. As compared with the previous studies, this study provides complex and systematic investigation in the field of LAS when the altered magnetic anisotropy is combined with the variable yield strength. Furthermore, this study also lists the suitable MBN parameters which can be easily employed for reliable distinction among LAS.

Reviewer: The introduction part is well-organized yet there is a reference problem. First, your reference list contains a few papers from polymers journal. If your work is convenient for this journal’s context then there are many references from this journal. Secondly, cited sources should be primary ones. Namely, indexed area shows the power of a paper and directly your paper’s reliability. Please make regulations in this direction.

Response: we do not understand this comment.

Manuscript: we prefer no change.

Reviewer: It is necessary to give quantitative and qualitative indicators of the proposed method in conclusion. Conclusions should be written in more detail adding numeric data. Conclusions section is inadequate. There should be the importance of the study in detail, comparison results with other approaches in literature, the success of the prediction and computational results.

Response: We improved the conclusions are required.

Manuscript: we added text

Applicability of different MBN parameters, for distinction of the LASs of the different yield strength, can be summarized as follows:

- MBN in the RD at the magnetising frequency of 750 Hz exhibits the systematic descent (90% decrease) and can be considered as the suitable MBN parameter, employed for this purpose (whereas the MBN in RD at the lower magnetising frequencies, as well as in TD, behave in a non-systematic manner),

- PP in the RD and TD at the lower magnetising frequencies (125 Hz and 250 Hz) are increasing along with the yield strength of LAS (50% increase for 125 Hz in RD and 40% increase in TD, 23% increase for 250 Hz in RD and 16% increase in TD); however, the PP should be combined with the number of MBN pulses (since at the higher magnetising frequencies the PP behave in a non-systematic manner),

- FWHM behaves in a non-systematic manner for all the magnetising conditions and directions,

- the number of all the MBN pulses, at the lower magnetising frequencies (especially 250 Hz), is increasing along with the yield strength of LAS (25% increase for 125 Hz in RD and 23% increase in TD, 37% increase for 250 Hz in RD and 30% increase in TD); however, the number of MBN pulses should be combined with the PP for the magnetising frequency of 125 Hz (the number of the MBN pulses at the higher magnetising frequencies behave in a non-systematic manner),

- applying the background threshold for counting of the MBN pulses number is not beneficial, in this particular case, with respect to materials characterisation.

Reviewer: Improve the results and discussion and conclusion parts.

Reviewer: It should be clearly stated the success rate of the study in abstract, results and discussion and conclusion parts.

Response: We altered all these parts. Please check mainly the texts highlighted in yellow.

Manuscript: Please check the revised version of manuscript and the corresponding parts. Revised as required.

Reviewer: There are typographical and eventual language problems in paper

Response: We had sent our manuscript to the native speaker to check the critical parts.

Manuscript: We went through the critical parts of our manuscript. Please check it.

Reviewer 2 Report

I read the article in detail. There are some important issues that need to be fixed. Acceptable after corrections.

1-There are serious grammatical errors in the abstract. In addition, the results obtained should be expressed clearly rather than general comments.

2- Parameters used for MBN can be specified in the introduction part. What parameters did you apply during the tests?

3-The regions seen in the figures shown in Figure.1 should be stated or marked. As such, the description of the pictures is insufficient.

4- In Figure.2, the differences in the graphics should be explained. Different colored curves at different mV values ​​should be interpreted in detail.

5- Comments for Figure.5 should be supported by literature studies.

6-In Figure.6 and figure 7, instantaneous changes are different from each other. For this reason, the explanation of the frequency effect in the comments is insufficient.

7- What exactly does the intersection point in Figure.8a mean, explained in detail?

8- The increasing or decreasing table created in Table 2 is meaningless. instead, the changes can be expressed as % or numerically.

9-Conclusion section should be expanded considering the effect of different parameters applied.

Author Response

Reviewer n. 2:

All changes made in the manuscript (additional texts and corrections) are highlighted yellow colour (valid for the manuscript as well as this document).

Reviewer: There are serious grammatical errors in the abstract.

Response: We had sent our manuscript to the native speaker to check the critical parts.

Manuscript: We went through the critical parts of our manuscript. Please check it.

Reviewer: In addition, the results obtained should be expressed clearly rather than general comments.

Response: We agree. We altered the text in the introduction. We added new text and we removed the general comments.

Manuscript:

… behaviour with respect to the direction of sheet rolling and the transversal direction can be found due to realignment of the domain walls. This study demonstrates that the position of Barkhausen noise envelopes and the number of Barkhausen noise pulses increase in a systematic manner at the lower magnetising frequencies and these parameters can be employed for distinction of the low alloyed steels investigated in this study. However, the increasing magnetising frequency makes attenuation of Barkhausen noise more remarkable for the low alloyed steels of the higher strength. Therefore, the effective value of Barkhausen noise at the magnetising frequency 750 Hz in the rolling direction exhibits the systematic descent along with the increasing yield strength and this parameter can be also used for distinction of the low alloyed steels after their thermomechanical processing. 

Reviewer: Parameters used for MBN can be specified in the introduction part. What parameters did you apply during the tests?

Response: We added further explanations as required (Introduction).

Manuscript:

MBN signal post-processing enables extraction of the different MBN parameters which can be employed for a proposal of suitable concepts when materials characterisation is required in a non-destructive manner. The effective value of MBN signal is a function of the number of detected pulses as well as their strength. These pulses can occur in the different range of magnetic fields with the different distribution. For this reason, also MBN envelopes can be employed for extraction of the peak position as the maxima of the MBN envelopes together with the width of this envelopes. The aforementioned MBN parameters are considered as the most common MBN features frequently integrated into scientific studies dealing with MBN (easily extracted, mostly provided by commercially available software) [31]. These parameters can be also linked with the physical processes associated with DWs motion which provides deeper insight into magnetisation of bodies [18, 22, 31].  

Reviewer: The regions seen in the figures shown in Figure.1 should be stated or marked. As such, the description of the pictures is insufficient.

Response: These figures contains scale at the right bottom to make idea about the size. But we added the information about the scanned area as required into the title of Figure 1.

Manuscript:

Figure 1. Metallographic images of investigated steels (scanned area approx. 0.238 mm2).

Reviewer: In Figure.2, the differences in the graphics should be explained. Different colored curves at different mV values should be interpreted in detail.

Response: each colour means the different LAS of the nominal yield strength. This information is provided using the legend in each plot. We only highlighted in the title of Figure 2 that the evolutions are plotted as a function of magnetising voltage (or frequency) and yield strength (means that the number in each plot is associated with the LAS of certain yield strength).

Detail explanation of especially the magnetising sweeps is provided in further text. We consider that this explanation is sufficient and provide physical interpretation of the differences among the different evolutions. Please check the text beyond the Figure 4 and before Figure 5.

Manuscript:

Figure 2. Voltage and frequency sweeps for LAS of variable yield strength.

Reviewer: Comments for Figure.5 should be supported by literature studies

Response: we added the corresponding references.

Manuscript:

One might argue that MBN dominates the nearby coercive force (nearby the zero magnetic field and the corresponding magnetising current) when the magnetic field is low [18, 24, 26], whereas the DW rotation dominates at a higher magnetic field [12, 13, 14], see Figure 5. For this reason, the amplitude of the magnetising field should not influence the MBN. However, this part of the study demonstrates that a certain fraction of DWs at lower magnetising fields really stays unpinned and dH/dt only affects the speed of the movable DWs [35]. 

Reviewer: In Figure.6 and figure 7, instantaneous changes are different from each other. For this reason, the explanation of the frequency effect in the comments is insufficient

Response: We agree. One hand, we explained the different evolution of MBN as well as the shape of MBN envelopes with respect of RD and TD. On the other hand, really the evolutions for the magnetising frequency 750 Hz is quite different as those for the lower magnetising frequencies. Therefore, we added explanation.

Manuscript:

It should be also noted that the evolution of MBN for the magnetising frequency 750 Hz differs from those for the lower magnetising frequencies in RD as well TD, see Figure 6. The MBN systematically decreases at the lowest magnetising frequency 750 Hz (see Figure 6a) since the fraction of DWs which are pinned in their position increases along with the increasing yield strength (also the superimposing contribution of DWs realignment of into TD takes certain role). On the other hand, it seems that the major fraction of DWs are unpinned and contributes to the MBN at the lower magnetising frequencies. The descending region for the magnetising frequency 750 Hz in TD is followed by the moderate grows for the LAS of the yield strength in the range from 500 to 960 MPa since the decreasing fraction of DWs in motion is compensated by the DWs realignment into TD (see Figure 6b).

Reviewer: What exactly does the intersection point in Figure.8a mean, explained in detail?

Response: We agree. This point is quite interesting and comment very valuable. We added further explanation.

Manuscript:

Figure 8 also illustrates that the PP is nearly unaffected by the magnetising frequency for LAS of the yield strength 960 MPa and this LAS represent the boundary when the ascending evolution of PP versus magnetising frequency is reversed by the increasing one. Also Figure 8b depicts that PP for this LAS are quite close. It can be therefore noted that the effects of the movable DWs fraction and their dynamics are balanced in this particular case.  

Reviewer: The increasing or decreasing table created in Table 2 is meaningless. instead, the changes can be expressed as % or numerically.

Response: We agree. We added numeral values reflect % ascent of descent of the MBN parameter - in order to quantify the evolutions. Please check the appearance of Table 2.

Manuscript: Please check the appearance of Table 2.

Reviewer: Conclusion section should be expanded considering the effect of different parameters applied.

Response: We summarised our findings as required in Conclusions.

Manuscript:

Applicability of different MBN parameters, for distinction of the LASs of the different yield strength, can be summarized as follows:

- MBN in the RD at the magnetising frequency of 750 Hz exhibits the systematic descent (90% decrease) and can be considered as the suitable MBN parameter, employed for this purpose (whereas the MBN in RD at the lower magnetising frequencies, as well as in TD, behave in a non-systematic manner),

- PP in the RD and TD at the lower magnetising frequencies (125 Hz and 250 Hz) are increasing along with the yield strength of LAS (50% increase for 125 Hz in RD and 40% increase in TD, 23% increase for 250 Hz in RD and 16% increase in TD); however, the PP should be combined with the number of MBN pulses (since at the higher magnetising frequencies the PP behave in a non-systematic manner),

- FWHM behaves in a non-systematic manner for all the magnetising conditions and directions,

- the number of all the MBN pulses, at the lower magnetising frequencies (especially 250 Hz), is increasing along with the yield strength of LAS (25% increase for 125 Hz in RD and 23% increase in TD, 37% increase for 250 Hz in RD and 30% increase in TD); however, the number of MBN pulses should be combined with the PP for the magnetising frequency of 125 Hz (the number of the MBN pulses at the higher magnetising frequencies behave in a non-systematic manner),

- applying the background threshold for counting of the MBN pulses number is not beneficial, in this particular case, with respect to materials characterisation.

Reviewer 3 Report

In the review of research article titled: Influence of magnetizing conditions on Barkhausen noise in Fe soft magnetic materials after thermo-mechanical treatment, the authors have described the research work very well covering a lot of aspects. I would like to see this article publish but after some minor modifications as follow;

1.      In abstract portion, authors have just given the general discussion about the research work. It is suggested to please highlight your achieved results (values) in the abstract.

2.      Introduction portion is also week, comprise of just general discussion, authors have not highlighted the previously reported results with specific parameters. i.e, its not good to describe ---- et al., have done this work. Its better to describe what they achieved from their work.

3.      Problem statement is missing in the introduction portion, what were drawbacks in previous study, which compelled the authors to carry out this study. Add the problem statement and also the highlight the work in application point of view.

4.      Please describe the significance of your work which deffers your work from previous literature in the last paragraph of introduction.

5.      English must be improved and manuscript should be revised by native English speaker.

Good luck

Author Response

All changes made in the manuscript (additional texts and corrections) are highlighted yellow colour (valid for the manuscript as well as this document).

Reviewer: In abstract portion, authors have just given the general discussion about the research work. It is suggested to please highlight your achieved results (values) in the abstract.

Response: We agree. We altered the text in the introduction. We added new text and we removed the general comments.

Manuscript:

… behaviour with respect to the direction of sheet rolling and the transversal direction can be found due to realignment of the domain walls. This study demonstrates that the position of Barkhausen noise envelopes and the number of Barkhausen noise pulses increase in a systematic manner at the lower magnetising frequencies and these parameters can be employed for distinction of the low alloyed steels investigated in this study. However, the increasing magnetising frequency makes attenuation of Barkhausen noise more remarkable for the low alloyed steels of the higher strength. Therefore, the effective value of Barkhausen noise at the magnetising frequency 750 Hz in the rolling direction exhibits the systematic descent along with the increasing yield strength and this parameter can be also used for distinction of the low alloyed steels after their thermomechanical processing. 

Reviewer: Introduction portion is also week, comprise of just general discussion, authors have not highlighted the previously reported results with specific parameters. i.e, its not good to describe ---- et al., have done this work. Its better to describe what they achieved from their work.

Response: We altered the text as required. We described directly the main findings of the cited documents.

We also added the brief discussion about the specific parameters linked with our topic – this task is revised in the last paragraph of Introduction as it is mentioned in the next item.

Manuscript:

Dychtoń et al. [15] linked increasing MBN with thermal softening after grinding and remarkable alteration of hardness in the thermally affected region. Neslušan et al. [16] investigated that the lower MBN after grinding can be found when carbides in martensite are unaffected and their decomposition is observed when MBN exceeds the critical threshold. Stupakov et al. [17] reported that the stronger MBN emission is received due to presence of decarburised surface layer of low dislocation density. Ktena et al. [18] found that MBN is increasing due to grain refinement as a result of increasing DWs density. Furthermore, the alignment of the DWs is affected by the stress state, and therefore, MBN can be employed for the measurement of stresses. Liu et al. [19] clearly proved that MBN grows along with tensile stresses as a result of DWs realignment. Kypris et al. [20] provided the model in which residual stress assessment can be obtained using MBN depth attenuation. Sorsa et al. [21] developed the regression model for prediction of residual stresses after shot peening. 

The MBN technique has already been used for monitoring LAS [22, 23] or assessing stresses in real highway bridges [6]. Pitoňák et al. [22] reported that MBN attenuates in LAS as results of variable Zn coating. Jančula et al. [23] found that MBN drops down along with increasing corrosion extent in LAS. Zgútová et al. [6] reported about sensitivity of the different MBN parameters with respect of tensile stress in LAS S460. Moreover, MBN as a function of strain hardening of S235 and tensile stress was investigated as well [24, 25]. Neslušan et al. [24] clearly demonstrated that MBN after the uniaxial plastic straining is decreasing in the direction of tensile stress whereas Jurkovič et al. [25] found remarkable non-homogeneity of plastic straining and the corresponding MBN after yielding. Blaow et al. [26] investigated the effect of LAS bending on MBN and reported about the decreasing MBN in the region of compressive stresses. Gutierez et al. [27] demonstrated the decreasing MBN as a result of reduced hardness and the corresponding dislocation density in AISI 4130. Kadavath et al. [28] also clearly proved that MBN drops down with the increasing hardness using a Jominy end-quenched test and the samples subjected to different cooling rates.

Reviewer: Problem statement is missing in the introduction portion, what were drawbacks in previous study, which compelled the authors to carry out this study. Add the problem statement and also the highlight the work in application point of view.

Reviewer: Please describe the significance of your work which deffers your work from previous literature in the last paragraph of introduction.

Response:

Manuscript:

Studies dealing with the influence of magnetising conditions and usage of the different MBN parameters for a specific purpose have been reported. The procedures for setting the optimised magnetising voltage and frequency can be found in [30]. Vashista and Moorthy [31] reported that the alterations in magnetic field affect appearance of MBN envelopes. It was also demonstrated that the position of MBN envelope can be easily linked with dislocation density and the corresponding hardness [32]. However, Blažek et al. [33] found that MBN envelope is also sensitive to the MBN signal filtration especially at the higher magnetising frequencies. Santa-aho et al. [34] employed the variable magnetising sweeps for detection of case-hardened depth. Santa-aho et al. [35] also executed features applicability such MBN, MBN envelopes and other features for prediction residual stresses in case-hardened bodies. Šrámek et al. [36] found that the width of MBN envelope drops down along with the increasing magnitude of tensile stresses. Also the number of MBN pulses was employed for deeper insight into magnetisation process of LAS. Jurkovič et al. [25] found that the refinement of microstructure after plastic deformation increases the number of detected pulses. These studies proved that proposal of suitable magnetising conditions is vital of importance in order to obtain acceptable sensitivity of MBN technique. Having in mind the diversity of LAS as well as a variety of magnetising conditions, multiple combinations of optimal magnetising conditions for LAS of different yield strengths can be found. This paper investigates this aspect in a systematic manner and explains the influence of the magnetising conditions on the acquired MBN signals. As compared with the previous studies, this study provides complex and systematic investigation in the field of LAS when the altered magnetic anisotropy is combined with the variable yield strength. Furthermore, this study also lists the suitable MBN parameters which can be easily employed for reliable distinction among LAS.

Reviewer: English must be improved and manuscript should be revised by native English speaker.

Response: We had sent our manuscript to the native speaker to check the critical parts.

Manuscript: We went through the critical parts of our manuscript. Please check it.